# Airway Ciliary Beating Affected by the *Pcp4* Dose-Dependent [Ca^2+^]_i_ Increase in Down Syndrome Mice, Ts1Rhr

**DOI:** 10.3390/ijms21061947

**Published:** 2020-03-12

**Authors:** Haruka Kogiso, Matthieu Raveau, Kazuhiro Yamakawa, Daichi Saito, Yukiko Ikeuchi, Tomonori Okazaki, Shinji Asano, Toshio Inui, Yoshinori Marunaka, Takashi Nakahari

**Affiliations:** 1Research Unit for Epithelial Physiology, Research Organization of Science and Technology, BKC, Ritsumeikan University, Kusatsu 525-8577, Japan; kogiso0520@gmail.com (H.K.); ph0104er@ed.ritsumei.ac.jp (D.S.); y.ikeuchi@quartz.ocn.ne.jp (Y.I.); ashinji@ph.ritsumei.ac.jp (S.A.); t-inui@saisei-mirai.or.jp (T.I.); marunaka@koto.kpu-m.ac.jp (Y.M.); 2Department of Molecular Cell Physiology, Graduate School of Medical Science, Kyoto Prefectural University of Medicine, Kyoto 602-8566, Japan; 3Laboratory for Neurogenetics, RIKEN, Brain Science Institute, Saitama 351-0198, Japan; matthieu.raveau@riken.jp (M.R.); yamakawa@med.nagoya-cu.ac.jp (K.Y.); 4Department of Neurodevelopmental Disorder Genetics, Institute of Brain Sciences, Nagoya City University Graduate School of Medical Sciences, Kawasumi, Mizuho-cho, Mizuho-ku Nagoya 467-8601, Japan; 5Department of Molecular Physiology, Faculty of Pharmaceutical Sciences, BKC, Ritsumeikan University, Kusatsu 525-8577, Japan; ph0034fe@ed.ritsumei.ac.jp; 6Saisei Mirai Clinics, Moriguchi 570-0012, Japan; 7Research Institute for Clinical Physiology, Kyoto Industrial Health Association, Kyoto 604-8472, Japan

**Keywords:** airway cilia, Down syndrome mouse, PDE1, intracellular Ca^2+^ concentration, cAMP, Pcp4, TRPV4

## Abstract

In Ts1Rhr, a Down syndrome model mouse, the airway ciliary beatings are impaired; that is, decreases in ciliary beat frequency (CBF) and ciliary bend angle (CBA, an index of ciliary beat amplitude)). A resumption to two copies of the *Pcp4* gene on the Ts1Rhr trisomic segment (Ts1Rhr:*Pcp4*^+/+/-^) rescues the decreases in CBF and CBA that occur in Ts1Rhr. In airway cilia, upon stimulation with procaterol (a β_2_-agonist), the CBF increase is slower over the time course than the CBA increase because of cAMP degradation by Ca^2+^/calmodulin-dependent phosphodiesterase 1 (PDE1) existing in the metabolon regulating CBF. In Ts1Rhr, procaterol-stimulated CBF increase was much slower over the time course than in the wild-type mouse (Wt) or Ts1Rhr:*Pcp4*^+/+/-^. However, in the presence of 8MmIBMX (8-methoxymethyl isobutylmethyl xanthine, an inhibitor of PDE1) or calmidazolium (an inhibitor of calmodulin), in both Wt and Ts1Rhr, procaterol stimulates CBF and CBA increases over a similar time course. Measurements of cAMP revealed that the cAMP contents were lower in Ts1Rhr than in Wt or in Ts1Rhr:*Pcp4*^+/+/-^, suggesting the activation of PDE1A that is present in Ts1Rhr airway cilia. Measurements of the intracellular Ca^2+^ concentration ([Ca^2+^]_i_) in airway ciliary cells revealed that temperature (increasing from 25 to 37 °C) or 4αPDD (a selective transient receptor potential vanilloid 4 (TRPV4) agonist) stimulates a larger [Ca^2+^]_i_ increase in Ts1Rhr than in Wt or Ts1Rhr:*Pcp4*^+/+/-^. In airway ciliary cells of Ts1Rhr, *Pcp4*-dose dependent activation of TRPV4 appears to induce an increase in the basal [Ca^2+^]_i_. In early embryonic day mice, a basal [Ca^2+^]_i_ increased by PCP4 expressed may affect axonemal regulatory complexes regulated by the Ca^2+^-signal in Ts1Rhr, leading to a decrease in the basal CBF and CBA of airway cilia.

## 1. Introduction

Down syndrome (DS) caused by trisomy of human chromosome 21 (HSA21) occurred in 14.5/10,000 live births in the USA between 2004 and 2006 [1,2]. DS patients have developmental delay, mental retardation, craniofacial and brain dysmorphology, and heart abnormalities. Moreover, DS patients have airway abnormalities, such as laryngomalacia, tracheomalacia, complete tracheal ring, and lingual tonsils [3,4], which cause recurrent and sustained infections in airways and lungs. Respiratory infections, such as bronchopneumonia, are frequent causes of death in DS children [4,5,6].

Congenital airway abnormalities may induce serious lung diseases in DS children [3,4]. Autopsy studies of DS have demonstrated that DS patients have enlarged alveoli and alveolar ducts in the lungs [7]. Moreover, in DS children, a reduced frequency and an abnormal waveform of nasal cilia have been reported, although the ciliary structures are not affected [8]. The reduced ciliary beat frequency (CBF) and abnormal waveform, such as a decrease in ciliary bend angle (CBA), decrease the rate of mucociliary clearance leading to recurrent and sustained airway infections [8,9]. 

There are many limitations in DS research, such as genotype–phenotype relationships, identification of dosage dependent genes, and testing therapeutic strategies [10,11,12]. To circumvent these limitations, DS mouse models with partial trisomy, Ts65DN, Ts1Cje, and Ts1Rhr, are widely used in research [10,11,12,13,14,15]. These mouse models are generated by targeting the chromosome 16 of mouse (MMU16) in its region of synteny to HSA21 [11,12,13,14,15]. Ts65DN, which harbors ~86 protein coding genes from *miR*155 to *Znf*295, is a large trisomy, Ts1Cje, which harbors ~66 protein coding genes from *Sfrs*15 to *Znf*295, and is shorter but overlaps with Ts65DN. Ts1Rhr, which harbors ~33 protein coding genes from *Cbr*3 to *Fam*3b, is the shortest trisomy and overlaps with Ts65DN and Ts1Cje. These three types of trisomy lead to DS-like learning and memory deficits and ventriculomegaly [12,14]. Ventriculomegaly has been shown to be caused by CBF and CBA decreases in ependymal ciliary beating [14,16,17]. Moreover, *Pcp*4 (Purkinje cell protein 4) localized within the Ts1Rhr region has also been demonstrated to play a crucial role in the maintenance of CBF and CBA in ependymal ciliary cells [14]. However, it remains uncertain whether dysfunction of airway cilia depends on *Pcp*4 in Ts1Rhr.

We measured the CBF and CBA of airway cilia in Ts1Rhr, using videomicroscopy equipped with a high-speed camera. In the course of the experiments, we found that, under basal conditions without any stimulation, the CBF and CBA of airway cilia were lower in Ts1Rhr than in Wt. A previous study demonstrated that *Pcp4* on the Ts1Rhr trisomic segment induces dose-dependent decreases in CBF and CBA in ependymal cilia and that resumption to two copies of *Pcp4* on Ts1Rhr trisomic segment (Ts1Rhr:*Pcp4*^+/+/-^) rescues the decreases in CBF and CBA [14]. However, in the airway cilia of Ts1Rhr, it remains uncertain whether *Pcp4* on the Ts1Rhr trisomic segment dose-dependently decreases the CBF and CBA of airway cilia. The goal of this study is to clarify the role of *Pcp4* on the Ts1Rhr trisomic segment in CBF and CBA of airway cilia. 

## 2. Results

### 2.1. CBF and CBA Under Unstimulated Conditions

Video images of the airway ciliary beating are shown in the Appendix A (A: Wt, B: Ts1Rhr, C: Ts1Rhr:*Pcp4^+/+/-^*). CBFs (160–320 cells/animal) and CBAs (30–70 cell/animal) of airway cilia were measured in Wt, Ts1Rhr, and Ts1Rhr:*Pcp4*^+/+/-^ [18,19,20,21]. Table 1 shows CBFs and CBAs (means ± SD) of airway cilia in Wt, Ts1Rhr, and Ts1Rhr:*Pcp4*^+/+/-^. CBF and CBA in airway cilia were significantly decreased in the Ts1Rhr compared with those in Wt (*p* < 0.01). However, in Ts1Rhr:*Pcp4*^+/+/-^, CBF and CBA were similar to those in Wt. Thus, the resumption of the *Pcp4* on the Ts1Rhr trisomic segment (Ts1Rhr:*Pcp4^+/+/-^*) rescues the decreases in CBF and CBA that occurred in Ts1Rhr (Figure 1).

### 2.2. Effects of Procaterol on CBF and CBA

#### 2.2.1. Effects of 1 nM Procaterol on CBF and CBA Increase in Wt, Ts1Rhr and Ts1Rhr:Pcp4^+/+/-^

Figure 2A shows that the CBF and CBA of airway cilia increased by stimulation with 1 nM procaterol in Wt. Stimulation with procaterol increased CBF and CBA, but the time course of the CBF increase was much slower than that of the CBA increase. Similar results have already been shown in previously reports [18,19,20,21]. Next, the same experiments were carried out using Ts1Rhr (Figure 2B). Stimulation with procaterol increased CBF and CBA in the airway cilia of Ts1Rhr, but the time course of the CBF increase was much slower in Ts1Rhr than in Wt; that is, CBF reached a plateau within 8 min from the start of procaterol stimulation, although it reached a plateau within 6 min from the start of procaterol stimulation in Wt. Experiments were also carried out using Ts1Rhr:*Pcp4*^+/+/-^ (Figure 2C). Stimulation with procaterol increased CBF and CBA in airway cilia of Ts1Rhr:*Pcp4*^+/+/-^. The time course of the CBF increase in Ts1Rhr:*Pcp4*^+/+/-^ was faster than that in Ts1Rhr. CBF reached a plateau within 6 min from the start of procaterol stimulation. Thus, in airway cilia of Ts1Rhr:*Pcp4*^+/+/-^, the time course of the CBF increase was similar to that in airway cilia of Wt. Table 2 shows CBF and CBA ratios 5 min from the start of procaterol stimulation in three types of mouse. The extents of the CBF and CBA increases stimulated by 1 nM procaterol were larger in Wt than in Ts1Rhr or Ts1Rhr:*Pcp4*^+/+/-^. The slower CBF increase in Ts1Rhr was recovered by the resumption of *Pcp4* on the Ts1Rhr trisomic segment (Ts1Rhr:*Pcp4*^+/+/-^). To quantitatively compare the time courses of CBF and CBA increases in Wt, Ts1Rhr, and Ts1Rhr:*Pcp4*^+/+/-^, the procaterol-stimulated increases in CBF and CBA were fitted to an exponential curve, a•[1-exp(-t/τ)] (where “a” is the value at t = ∞ in each experiment, “τ” is the time from the start of procaterol stimulation, and “τ” is the time constant in the change of CBF or CBA). “τ” was used to assess the time course of the CBF or CBA increase stimulated by procaterol. Figure 2D shows the normalized values of CBA and CBF (%) and the fitted curves in three experiments. The normalized values of the CBF and CBA increased were calculated as (CBF_t_–CBF_0_)/(CBF_∞_–CBF_0_) and (CBA_t_–CBA_0_)/(CBA_∞_–CBA_0_) (where the subscripts “t”, “0” and “∞” show the time from the start of procaterol stimulation). The values of CBF_∞_ and CBA_∞_ were calculated by applying the equation, a•[1-exp(-t/τ)], to the observed data for CBF and CBA. “τ” for the procaterol-stimulated CBF increase were summarized in Table 3. “τ” in Ts1Rhr was significantly larger than that in Wt and Ts1Rhr:*Pcp4*^+/+/-^ (*p* < 0.05). Thus, τ in Ts1Rhr:*Pcp4*^+/+/-^ recovered to the value found in Wt. Unlike CBF, the values of τ for the CBA increase stimulated by procaterol were similar in the three genotypes of mouse. 

Previous studies in *Chlamydomonas* and *Tetrahymena* mutants and in primary ciliary dyskinesia (PCD) showed that the outer dynein arms (ODAs) and inner dynein arms (IDAs) are functionally distinct: The ODAs control CBF, and the IDAs control the waveform including CBA [22,23,24,25]. Moreover, during procaterol stimulation, the slower CBF increase than CBA increase has been shown to be caused by Ca^2+^/calmodulin dependent phosphodiesterase 1A (PDE1A), which exists in the metabolon-regulating ODAs (outer dynein arms) and degrades cAMP dependent on [Ca^2+^]_i_ [19,20,21]. These results suggest that accumulations of cAMP in the metabolon-regulating ODAs are slow in Ts1Rhr compared with those in Wt and Ts1Rhr:*Pcp4^+/+/-^*. The activity of PDE1A localized in the metabolon regulating the ODA of airway cilia may be higher in Ts1Rhr than in Wt or Ts1Rhr:*Pcp4*^+/+/-^, suggesting a high [Ca^2+^]_i_ in the metabolon-regulating ODAs.

#### 2.2.2. Effects of 10 nM Procaterol on CBF and CBA Increase in Wt, Ts1Rhr and Ts1Rhr:Pcp4^+/+/-^

Similar experiments were carried out using 10 nM procaterol. Previous studies have demonstrated that procaterol at 10 nM accumulates a larger amount of cAMP in airway ciliary cells than that at 1 nM [19]. Stimulation with 10 nM procaterol largely increased CBF and CBF in Wt (Figure 3A), and CBF increase was slower in the time course than CBA increase [18,19]. The CBF and CBA ratios and τ for CBF increase during 10 nM procaterol stimulation are shown in Table 2 and Table 3. Unlike stimulation with 1 nM procaterol, stimulation with 10 nM procaterol increased CBF and CBA in Ts1Rhr, similar to those in Wt (Figure 3B, Table 2). τs for the CBF increase stimulated by 10 nM procaterol were similar in Ts1Rhr and Wt (Table 3). Thus, upon accumulation of a large amount of cAMP, the extents and the time courses of CBF and CBA increases are similar in both Ts1Rhr and Wt, suggesting that PKA signals increasing CBF and CBA are not affected in Ts1Rhr.

### 2.3. Effects of PDE1 Inhibition on Procaterol-Stimulated CBF and CBA

#### 2.3.1. Effects of 8MmIBMX on the CBF and CBA Increases Stimulated by 1 nM Procaterol

The effects of 8MmIBMX (a selective PDE1 inhibitor) on the procaterol-stimulated CBF and CBA increases were examined in airway cilia of Wt and Ts1Rhr. Previous studies demonstrated that 8MmIBMX at 40 µM inhibits PDE1A in airway ciliary cells [19,20,21]. In Wt, the addition of 8MmIBMX (40 µM) alone increased CBF and CBA of airway cilia. Further stimulation with procaterol increased CBF and CBA over a similar time course, which reached plateaus within 4 min (Figure 4A). The CBF and CBA ratios are summarized in Table 4. Comparing the time courses of the procaterol-stimulated CBF and CBA increases between airway cilia of Wt and Ts1Rhr, τ was calculated by the curve fitting. In the presence of 8MmIBMX, the time courses of the procaterol-stimulated CBF and CBA increases in Wt were similar to those in Ts1Rhr (Figure 4C). τs for the procaterol-stimulated CBF and CBA increases were similar in both Wt and Ts1Rhr (Table 3).

#### 2.3.2. Effects of Calmidazolium on the CBF and CBA Increases Stimulated by 1 nM Procaterol

Similar experiments were carried out using calmidazolium (25 µM, an inhibitor of calmodulin), instead of 8MmIBMX, because PDE1 is a Ca^2+^/calmodulin dependent enzyme. Previous studies demonstrated that calmidazolium at 25 µM inhibits PDE1A in airway ciliary cells [19,20,21]. The addition of calmidazolium (25 µM) and further stimulation with 1 nM procaterol increased CBF and CBA to the similar level with similar time courses in Wt and Ts1Rhr (Figure 5). Thus, the results obtained from the calmidazolium experiments (Figure 5) were similar to those obtained from the 8MmIBMX experiments (Figure 4). The CBF and CBA ratios in the presence of calmidazolium addition were summarized in Table 4. The time courses of the procaterol-stimulated CBF and CBA increases (τs for their increases) were similar in Wt and Ts1Rhr (Table 3).

### 2.4. Cyclic AMP Contents in Isolated Lung Cells

Figure 6 shows the cAMP contents of lung cells isolated from Wt, Ts1Rhr, and Ts1Rhr:Pcp4^+/+/-^. The cells were incubated in the control solution for 20 min at 37 °C. The contents of cAMP were 40.8 ± 1.0 (*n* = 3 animals), 32.9 ± 1.0 (*n* = 3 animals), and 42.7 ± 1.3 (*n* = 3 animals) nmoles/g-protein in cells from Wt, Ts1Rhr, and Ts1Rhr:*Pcp4*^+/+/-^, respectively. Thus, under unstimulated conditions, the cAMP contents of lung cells from Ts1Rhr were lower than those of lung cells from Wt. However, cAMP content in Ts1Rhr:*Pcp4*^+/+/-^ was similar to that in Wt. As shown in Figure 2 and Figure 4, Figure 5 and Figure 6, cAMP degradation via PDE1 was greater in airway ciliary cells of Ts1Rhr than in airway ciliary cells of Wt or Ts1Rhr:*Pcp4*^+/+/-^. These results suggest that the basal [Ca^2+^]_i_ in airway ciliary cells of Ts1Rhr may be higher than that of Wt or Ts1Rhr:*Pcp4*^+/+/-^. A high [Ca^2+^]_i_ activates PDE1A leading to a decrease in the cAMP content in airway ciliary cells.

### 2.5. [Ca^2+^]_i_ of Airway Ciliary Cells Stimulated by Temperature and 4αPDD

In airway ciliary cells, transient receptor potential vanilloid 4 (TRPV4) channels participate in the maintenance of the [Ca^2+^]_i_ under an unstimulated condition [26,27,28]. The effects of temperature elevation from 25 °C to 37 °C and 4α-Phorbol 12,13-didecanoate (4αPDD, a selective agonist for TRPV4) on the [Ca^2+^]_i_ were examined in airway ciliary cells from Wt, Ts1Rhr, and Ts1Rhr: *Pcp4*^+/+/-^. The changes in the [Ca^2+^]_i_ of airway ciliary cells were monitored by the fura 2 fluorescence ratio (F340/F380). Upon increasing the temperature from 25 °C to 37 °C, F340/F380 immediately increased in airway ciliary cells from Wt. In Ts1Rhr, the temperature increase also increased F340/F340, and the extents of F340/F380 increase were enhanced in airway ciliary cells from Ts1Rhr. However, in airway ciliary cells from Ts1Rhr:*Pcp4*^+/+/-^, the high temperature increased F340/F380, but the extent of the F340/F380 increase was similar to that in cells from Wt (Figure 7A). The increases in F340/F380 over the course of a 10 min temperature increase are summarized in Figure 7B. The F340/F380 increased in airway ciliary cells from Ts1Rhr were significantly larger than that in airway ciliary cells from Wt or Ts1Rhr:*Pcp4*^+/+/-^ (paired Student’s t test). The increases of F340/F380 of airway ciliary cells from the three genotypes of mice were also measured during stimulation with 4αPDD. Stimulation with 4αPDD slightly, but significantly, increased F340/F380 at 37 °C (paired Student’s t test). The increases in F340/F380 10 min after 4αPDD stimulation are summarized in Figure 7B. The increases of F340/F380 in airway ciliary cells stimulated by 4αPDD were significantly larger in Ts1Rhr than in Wt or Ts1Rhr:*Pcp4*^+/+/-^ (paired Student’s t test).

## 3. Discussion

The present study demonstrated that the basal CBF and CBA of airway cilia decrease in Ts1Rhr, but not in Ts1Rhr:*Pcp4*^+/+/-^. This result indicates that resumption to two copies of the *Pcp4* gene on the Ts1Rhr trisomic segment (Ts1Rhr:*Pcp4*^+/+/-^) rescues the decreases in the basal CBF and CBA of airway cilia that occurred in Ts1Rhr. A *Pcp4* dose-dependent ciliary dysfunction in ependymal cilia (decreases in basal CBF and CBA) has already been reported in Ts1Rhr [14]. Three copies of the *Pcp4* gene induce overexpression of its transcripts and proteins in a model mouse of DS, Ts1Cje [29]. A CBF decrease and an abnormal waveform under basal conditions have already been shown in human nasal ciliary cells from DS patients [8]. These results suggest that an overexpression of PCP4 in Ts1Rhr induces dysfunction of the airway ciliary beating.

The present study also demonstrated that Ca^2+^ entry in airway ciliary cells stimulated by temperature is enhanced in Ts1Rhr compared with that in Wt. Previous studies have shown that airway ciliary cells express TRPV4 channels activated by temperature or 4αPDD, and that the TRPV4 channels participate in the maintenance of the basal [Ca^2+^]_i_ in airway ciliary cells [26,27]. 

The ODAs and IDAs are different in their functions: The ODAs control CBF, and the IDAs control the waveform including CBA [22,23,24,25]. Previous studies have demonstrated that PDE1A exists in the metabolon regulating ODAs and that an increase in the [Ca^2+^]_i_ degrades cAMP in the metabolon regulating ODAs in airway cilia leading to a delay in the time course of the CBF increase stimulated by procaterol [19,20,21]. The present study clearly shows that the time course of the CBF increase stimulated by 1 nM procaterol in airway cilia is slower in Ts1Rhr than in Wt, and that the slower CBF increase in Ts1Rhr is caused by the activation of PDE1A [19,20,21]. These observations suggest that the basal [Ca^2+^]_i_ may be higher in airway ciliary cells of Ts1Rhr than in airway ciliary cells of Wt. The high [Ca^2+^]_i_ in airway ciliary cells of Ts1Rhr appears to be induced by an enhancement of Ca^2+^ entry via TRPV4.

On the other hand, the slower procaterol-stimulated CBF increase in airway cilia of Ts1Rhr was not observed in airway cilia of Ts1Rhr:*Pcp4^+/+/-^*. Moreover, the enhancement of Ca^2+^ entry stimulated by temperature or 4αPDD occurred in Ts1Rhr was not noted in Ts1Rhr:*Pcp4*^+/+/-^. PCP4, a regulator of Ca^2+^ signaling, modulates calmodulin activity by activating the interaction of Ca^2+^ and calmodulin [29,30,31,32,33]. Previous studies have demonstrated that PCP4 overexpression increases the expression of CaMKII in the cerebellum [29,30] and PC12 cells [34], which may activate calmodulin activity. Moreover, this study suggests that PCP4 overexpression may affect TRPV4 expression in airway ciliary cells leading to a high basal [Ca^2+^]_i_, which activates calmodulin activities. An increase in the [Ca^2+^]_I_, at least, enhances the interactions between Ca^2+^ and calmodulin modulated by PCP4, although the expression level of TRPV4 in Ts1Rhr is currently unknown. 

This study demonstrated that the basal CBF and CBA in Ts1Rhr are lower than those in Wt. Similar decreases in CBA and CBF have already been shown in ependymal cilia of Ts1Rhr. In ependymal cilia of Ts1Rhr:*Pcp4*^+/+/-^, the decreases in the basal CBF and CBA that occurred in Ts1Rhr are rescued, as shown in airway cilia of Ts1Rhr:*Pcp4*^+/+/-^. An increase in the [Ca^2+^]_i_ is known to increase CBF and CBA in airway ciliary cells [9,35,36]. The results of this study showing decreases in CBF and CBA in Ts1Rhr are inconsistent with previous observations indicating that an increase in the [Ca^2+^]_i_ increases CBF and CBA in Wt airway cilia [35]. The [Ca^2+^]_i_ in airway ciliary cells of Ts1Rhr:*Pcp4*^+/+/-^ is similar to that of Wt. The mutant genes associated with primary ciliary dyskinesia (PCD) have been shown to encode proteins, which are involved in axonemal structural and functional components, regulatory complexes, and ciliary assembly complexes and the PCD mutations affect the ultrastructure as detected by electron microscopy [37]. In a PCD model mouse, mutation of the outer dynein arm heavy chain 11 locus caused no ciliary beating [38]. However, in nasal cilia of DS patients, the ultrastructural examination by electronmicroscopy have revealed a normal architecture of cilia, although they show a reduced CBFs and abnormal wave forms [8]. PCP4, which participates in the functional properties of the calcium-calmodulin signal, is detected in early embryonic days, and its overexpression induces abnormal neuronal development [25,26]. Ca^2+^-calmodulin signaling in airway ciliary cells activated by PCP4 overexpression on early embryonic days may affect the axonemal regulatory complexes regulated by Ca^2+^-signal, such as CaMKII binding complex, not the ultrastructure of cilia. 

The final levels of CBF and CBA stimulated by 1 nM procaterol in Ts1Rhr and Ts1Rhr:*Pcp4*^+/+/-^ were lower than those in Wt, although the final levels of CBF and CBA in Ts1Rhr stimulated by 10 nM procaterol were similar to those in Wt. A high level of cAMP accumulation stimulated by 10 nM procaterol can fully increase CBF and CBA in Ts1Rhr, but a moderate level of AMP accumulation stimulated by 1 nM procaterol may partially increase CBF and CBA. Ts1Rhr, although the shortest trisomy, still harbors ~32 protein coding genes from *Cbr*3 to *Fam*3b except Pcp4. These genes may affect axonemal regulatory complexes regulated by the PKA signaling pathway, leading to a low CBF and CBA levels stimulated by procaterol.

In DS children, respiratory diseases, such as bronchopneumonia, are the main cause of death [5,6], and they suffer from recurrent and prolonged upper respiratory infections. DS children have congenital abnormalities of their airways, heart, and immune system, which induce frequent respiratory infections, such as bronchopneumonia. Moreover, dysfunction of ciliary beating also induces recurrent and prolonged infections of airways and tends to be worse in lung diseases compared with that in typical patients [9,35,36].

The result of this study is summarized in Figure 8. Three copies of *Pcp4* on Ts1Rhr stimulate TRPV4 expression leading to increase basal [Ca^2+^]_i_. A basal [Ca^2+^]_i_ increase on early embryonic days may affect axonemal regulatory complexes regulated by Ca^2+^ signals and it also affects airway ciliary beating via interactions between cAMP and Ca^2+^ signals, such as PDE1A activation in Ts1Rhr. 

## 4. Materials and Methods

### 4.1. Mouse Lines and Genotyping

Dp(16*Cbr1*-*ORF9*)1Rhr, referred to as Ts1Rhr, was obtained from Jackson Laboratory (www.jax.org; date last accessed 12 January 2017, Stock numbers 005383 and 013530, respectively) and was maintained by crossing carrier males with C57BL/6J females. Ts1Rhr PCR genotyping was performed as described by the Jackson Laboratory. The genotypes of the *Pcp4* knockout mice were determined using the following specific PCR primers: Pcp4-R1: GCTGCACTTAGGCACAAATC; Pcp4-KO-F2: AGCAACAG GTTTCCTTGTGG; Pcp4-WT-F2: GAAT GCCT CTCATTGGTTGG. These primers amplified PCR products of 596 bp for the WT allele and 423 bp for the mutant allele, as previously reported [14].

### 4.2. Generation of Pcp4 Knockout Mice

*Pcp4* heterozygous knockout mice (*Pcp4*^+/-^) were obtained by using in-vivo Cre/loxP recombination [14]. Briefly, we inserted loxP sites upstream and downstream of *Pcp4* exon one. Heterozygous flox-neo females were mated to Tg(*EIIa-cre*)C5379Lmgd males [39] resulting in the deletion of both *Pcp4*exon one and neomycin resistance gene. Amplification and sequencing of a 423bp fragment overlapping the *Pcp4* exon one confirmed its deletion [39].

### 4.3. Solution and Chemicals

The control solution contained (in mM): 121 NaCl; 4.5 KCl; 25 NaHCO_3_; 1 MgCl_2_; 1.5 CaCl_2_; 5 Na-HEPES; 5 H-HEPES; and 5 glucose, and was aerated with 95% O_2_ and 5% CO_2_. The pH of solution was adjusted to 7.4 by adding 1N-HCl. The experiments were carried out at 37 °C. DNase I and bovine serum albumin (BSA) were purchased from Sigma (St. Louis, Mo, USA), procaterol, calmidazolium, 8-methoxymethyl-IBMX (8MmIBMX, a selective inhibitor of phosphodiesterase 1 (PDE1)), 4α-Phorbol 12,13-didecanoate (4αPDD), dimethyl sulfoxide (DMSO), and elastase were purchased from Wako Pure Chemical Industries, Ltd. (Osaka, Japan); and, Nembutal was purchased from Dainippon-Sumitomo Pharmaceutical Co Ltd (Tokyo). All reagents were dissolved in DMSO and prepared to their final concentrations immediately before the experiments. The DMSO concentration did not exceed 0.1%, and DMSO at this concentration had no effect on CBF and CBA [19,20,21,40].

### 4.4. Cell Preparation

Airway ciliary cells were isolated from the lungs as previously described [18,19,20,21,40]. Briefly, mice were anesthetized by inhalational isoflurane (3%), followed by intraperitoneal injections (ip) of pentobarbital sodium (70 mg/kg, Nembutal) and heparin (1000 units/kg), and heparinization for 15 min. Then, mice were sacrificed by a high-dose of pentobarbital sodium (100 mg/kg, ip). After the sacrifice, lungs were cleared of blood by perfusion via the pulmonary artery, and the lungs with the trachea and the heart were removed from the animal en bloc. The lung cavity was washed four times with a nominally Ca^2+^-free solution (0.5 mL) via the tracheal cannula. In the final wash, the Ca^2+^-free solution (0.5 mL) was retained in the lung cavity for 5 min, and the lung cavity was then washed five times with the control solution via the tracheal cannula. Finally, the control solution containing elastase (0.2 mg/mL) and DNase I (0.02 mg/mL) was instilled into the lung cavity and the airway epithelium was digested for 40 min at 37 °C. Following this incubation, the lungs were minced using fine forceps in the control solution containing DNase I (0.02 mg/mL) and BSA (5%). The minced tissue was filtered through a nylon mesh (a sieve having 300 µm openings). The cells were washed three times with centrifugation (160× *g* for 5 min) and then suspended in the control solution. The cell suspension was stored at 4 °C and cells were used within 5 h after the isolation.

### 4.5. Ethical Approval

The procedures and protocols for the experiments are approved by the Animal Research Committee of Kyoto Prefectural University of Medicine (No. 26-254, 6 November 2015) and Ritsumeikan University (BKC2018-024 and H3015 27 July 2018), and the experiments were performed in accordance with the guidelines of this committee.

### 4.6. CBA and CBF Measurements

Cells were placed on a coverslip precoated with Cell-Tak (Becton Dickinson Labware, Bedford, MA, USA). The coverslips were placed in a microperfusion chamber (20 μL) mounted on an inverted light microscope (Eclipse Ti, NIKON, Tokyo, Japan) connected to a high-speed camera (IDP Express R2000, Photron Ltd., Tokyo, Japan). The stage of the microscope was heated to 37 °C, since CBF is highly dependent on temperature [41]. Cells were perfused at 200 μL/min with the control solution aerated with a gas mixture (95% O_2_ and 5% CO_2_) at 37 °C. Ciliary cells, which were distinguished from other cells by their beating cilia, accounted for 10%–20% of isolated lung cells. For the CBA (angle) and CBF measurements, video images were recorded for 2 s at 500 fps [18,19,20,21]. Before the start of the experiments, the cells were perfused with the control solution for 5 min. After the experiments, CBA and CBF were measured using an image analysis program (DippMotion 2D, Ditect, Tokyo, Japan). The method to measure CBA and CBF has been previously described in detail [18,42,43]. The CBA and CBF ratios (CBA_t_/CBA_0_ and CBF_t_/CBF_0_), the values of which were normalized by the control values, were used for comparisons among the experiments. CBA or CBF was measured every 1 min during control perfusion (5 min) and was then averaged and the average value was used as CBA_0_ or CBF_0_. The subscripts “0” and “t” indicate the time before or after the start of experiments, respectively. Each experiment was carried out using 4–10 cover slips with cells obtained from 2–5 animals. From each coverslip, we selected 1–2 cells or a cell block and measured their CBAs and CBFs. The normalized CBA and CBF (CBA ratio and CBF ratio) calculated from 4–12 cells were plotted and “n” represents the number of cells.

### 4.7. Monitoring of the Intracellular Ca^2+^ Concentration ([Ca^2+^]_i_)

The [Ca^2+^]_i_ was monitored using fura 2 fluorescence (Ca^2+^ fluorescence dye). Isolated ciliary cells were incubated with 2.5 mM fura 2-AM (Dojindo, Kumamoto) for 30 min at 37 °C. Fura 2 fluorescence was measured using a fluorescence image analysis system (MetaFlour, Molecular Devices Japan). Fura 2 was excited at 340 and 380 nm, and emission was measured at 510 nm. The ratio of the fluorescence intensity (F_340_/F_380_) was calculated. The subscripts “0” and “t” indicate the time before or after the start of experiments, respectively.

### 4.8. Measurement of cAMP Contents 

Isolated lung cells were incubated in a control solution for 15 min at 37 °C. After the control incubation, the cells were immediately frozen in the liquid nitrogen and stored at −80 °C until the cAMP measurements. The protein and cAMP contents in the cells were measured using a Pierce BCA Protein Assay kit (Thermo Fisher Scientific K.K, Yokohama, Japan) and a cAMP EIA kit (Cayman Chemical, Ann Arbor, MI, USA), respectively. The cAMP contents were expressed as μmoles/gram-protein.

### 4.9. Statistical Analysis

Data were expressed as means ± SD or SEM. Statistical significance was assessed by analysis of variance (ANOVA) or Student’s *t*-test, as appropriate. Differences were considered significant at *p* < 0.05.

## 5. Conclusions

In Ts1Rhr, PCP4 overexpressed by three copies of *Pcp4* on Ts1Rhr stimulate an increase in the basal [Ca^2+^]_i_ via TRPV4. This basal [Ca^2+^]_i_ increase affects airway ciliary beating in Ts1Rhr, which appears to induce the ciliary dysfunction in airways. The ciliary dysfunction may be worse in lung diseases occurred in DS. 

## Figures and Tables

**Figure 1 ijms-21-01947-f001:**
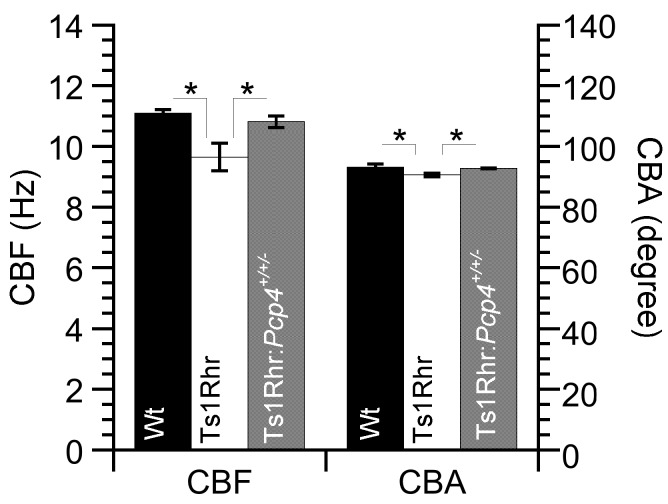
Basal ciliary beat frequency (CBF) and ciliary bend angle (CBA) of airway cilia in wild type (Wt), Ts1Rhr, and Ts1Rhr:*Pcp4*^+/+/-^. The basal CBF and CBA of airway cilia are lower in Ts1Rhr than in Wt. However, in Ts1Rhr:*Pcp4*^+/+/-^, they are similar to those in Wt. * significantly different (*p* < 0.05).

**Figure 2 ijms-21-01947-f002:**
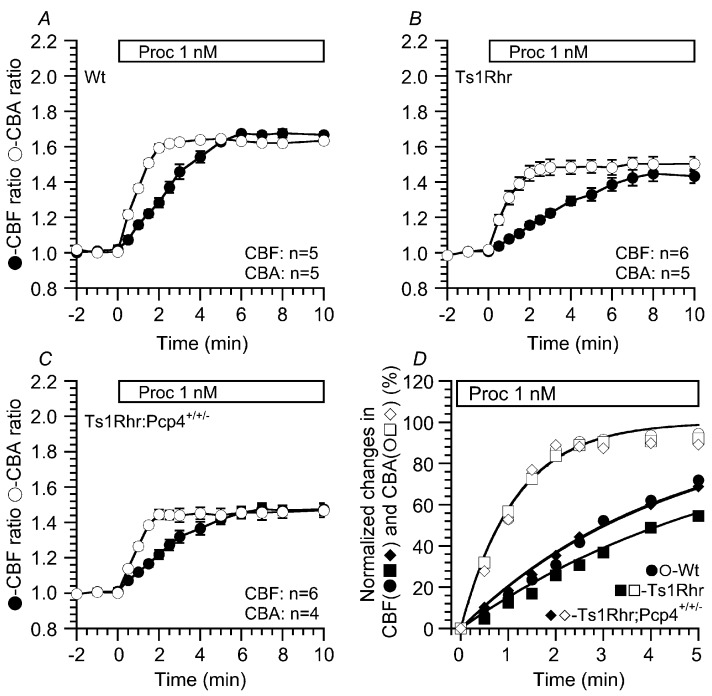
Increases in CBF and CBA ratios of airway cilia stimulated by 1 nM procaterol. Stimulation with procaterol (1 nM) increases CBF and CBA ratios of airway cilia in Wt, Ts1Rhr, and Ts1Rhr:*Pcp4*^+/+/-^. (**A**) Wt. The time course of the procaterol-stimulated CBF-increase is slower than that of the procaterol-stimulated CBA-increase. (**B**) Ts1Rhr. The time course of the procaterol-stimulated CBF increase is much slower in Ts1Rhr than in Wt. The plateau levels of the procaterol-stimulated CBF and CBA increases in Ts1Rhr are lower than those in Wt. (**C**) Ts1Rhr:*Pcp4*^+/+/-^. The time course of the procaterol stimulated CBF increase in Ts1Rhr:*Pcp4*^+/+/-^ is similar to that in Wt, although the plateau levels of CBF and CBA during procaterol stimulation are lower in Ts1Rhr than in Wt. (**D**) The time courses of procaterol-stimulated CBF and CBA increases in Wt, Ts1Rhr, and Ts1Rhr:*Pcp4*^+/+/-^. The procaterol-stimulated increases in CBF and CBA (normalized values) were fitted to an exponential curve, a•[1-exp(-t/τ)]. The fitted curves show a slower time course of the procaterol-stimulated CBF increase in Ts1Rhr than that in Wt or Ts1Rhr:*Pcp4*^+/+/-^, although the time courses of the CBA increases are similar in Wt, Ts1Rhr, and Ts1Rhr:*Pcp4*^+/+/-^.

**Figure 3 ijms-21-01947-f003:**
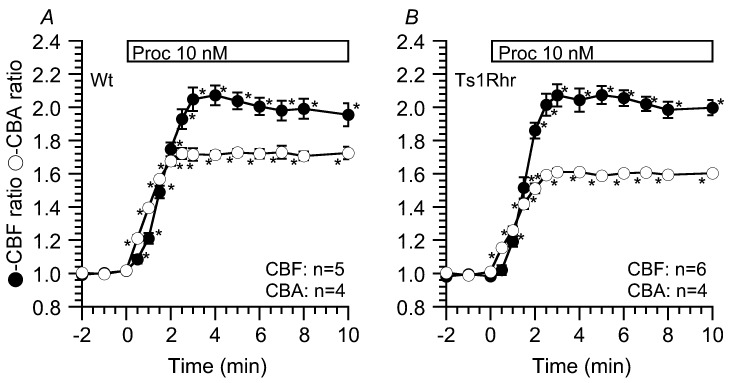
Stimulation with procaterol (10 nM) similarly increased the CBA and CBF ratios in airway cilia of Wt (**A**) and Ts1Rhr (**B**). The time course of the procaterol-stimulated CBF increase is slower than that of the CBA increase, and the plateau levels of the CBF and CBA increases by procaterol stimulation in Ts1Rhr are also similar to those in Wt. * significantly different vs values before stimulation (*p* < 0.05)

**Figure 4 ijms-21-01947-f004:**
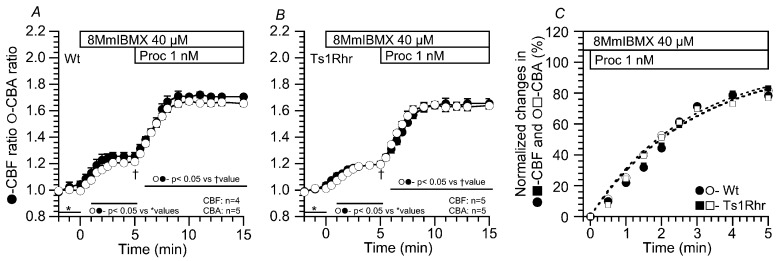
Effects of 8MmIBMX on the procaterol-stimulated CBF and CBA increases. The addition of 8MmIBMX (40 µM) increased CBF and CBA. (**A**) Wt. In the presence of 8MmIBMX, 1 nM procaterol stimulation increased the CBF and CBA of airway cilia over a similar time course. (**B**) Ts1Rhr. In the presence of 8MmIBMX, 1 nM procaterol stimulation increased CBF and CBA over a similar time course. The time courses of CBF and CBA increases in airway cilia of Ts1Rhr are similar to those in airway cilia of Wt. (**C**) The time courses of procaterol-stimulated CBF-increase and CBA-increase in Wt- and Ts1Rhr-airway cilia. The procaterol-stimulated increases in CBF and CBA (normalized values) were fitted to an exponential curve, a•[1-exp(-t/τ)]. The fitted curves show that the time courses of the procaterol-stimulated CBF and CBA increases of airway cilia are similar in Wt and Ts1Rhr.

**Figure 5 ijms-21-01947-f005:**
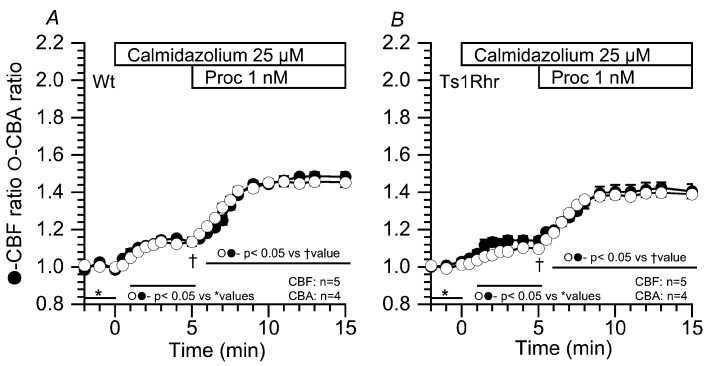
Effects of calmidazolium on the procaterol-stimulated CBF and CBA increases. The addition of calmidazolium (25 µM) increased CBF and CBA. (**A**) Wt. In the presence of calmidazolium, stimulation with 1 nM procaterol increased the CBF and CBA of airway cilia over a similar time course. (**B**) Ts1Rhr. In the presence of calmidazolium, stimulation with 1 nM procaterol increased the CBF and CBA over a similar time course. The time courses and the extents of the CBF and CBA increases in airway cilia of Ts1Rhr are similar to those in airway cilia of Wt.

**Figure 6 ijms-21-01947-f006:**
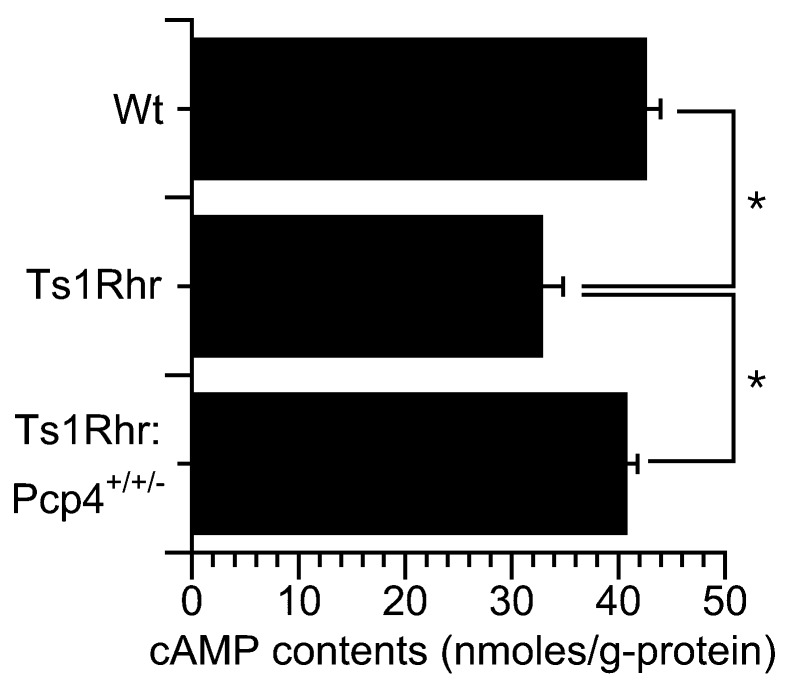
Cyclic AMP (Adenosine monophosphate) contents in isolated lung cells from Wt, Ts1Rhr, and Ts1Rhr:*Pcp4*^+/+/-^. Under unstimulated conditions, the cAMP content of lung cells from Ts1Rhr was lower than that of lung cells from Wt or Ts1Rhr:*Pcp4*^+/+/-^. * significantly different (*p* < 0.05)

**Figure 7 ijms-21-01947-f007:**
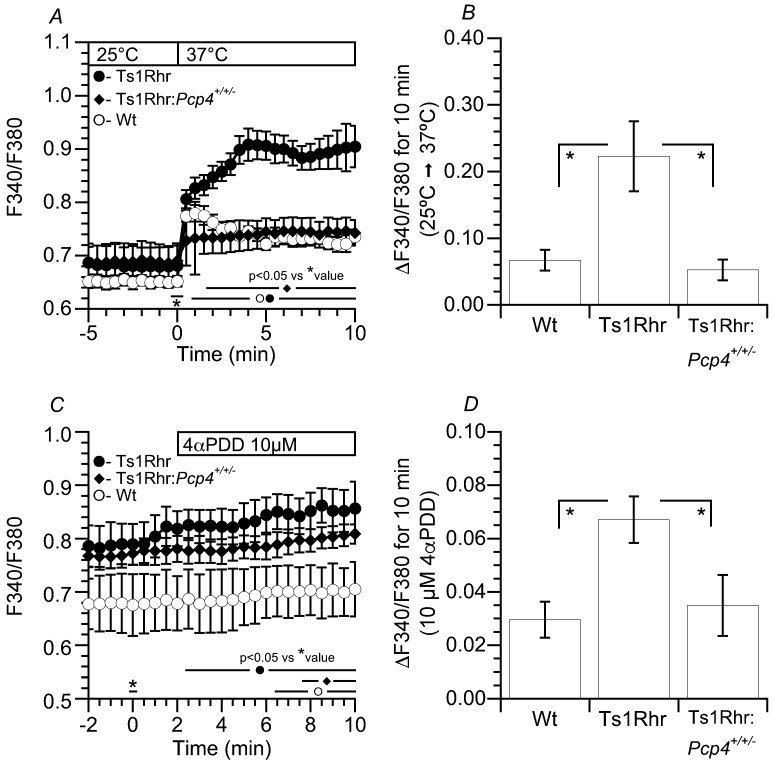
Changes in the [Ca^2+^]_i_ in airway ciliary cells stimulated by temperature and 4αPDD in Wt, Ts1Rhr, and Ts1Rhr:*Pcp4*^+/+/-^. (**A, B**) Temperature. (**A**) Increasing the temperature from 25 °C to 37 °C increased F340/F380 in cells from Wt. The temperature-stimulated increase in F340/F380 was enhanced in airway ciliary cells from Ts1Rhr, but it was similar in those from Ts1Rhr:*Pcp4*^+/+/-^. (**B**) Increases in F340/F380 (ΔF340/F380) 10 min after the temperature increase. ΔF340/F380 in cells from Ts1Rhr is significantly higher than that from Wt or from Ts1Rhr:*Pcp4*^+/+/-^. (**C, D**) 4αPDD (a selective agonist of TRPV4). (**C**) Stimulation with 4αPDD slightly increased F340/F380 in Wt at 37 °C. The 4αPDD-stimulated increase in F340/F380 was enhanced in cells from Ts1Rhr, but was similar in cells from Ts1Rhr:*Pcp4*^+/+/-^. (**D**) Increases in F340/F380 (ΔF340/F380) 10 min after 4αPDD stimulation. ΔF340/F380 in cells from Ts1Rhr is significantly higher than that from Wt or Ts1Rhr:*Pcp4*^+/+/-^. * significantly different (*p* < 0.05)

**Figure 8 ijms-21-01947-f008:**
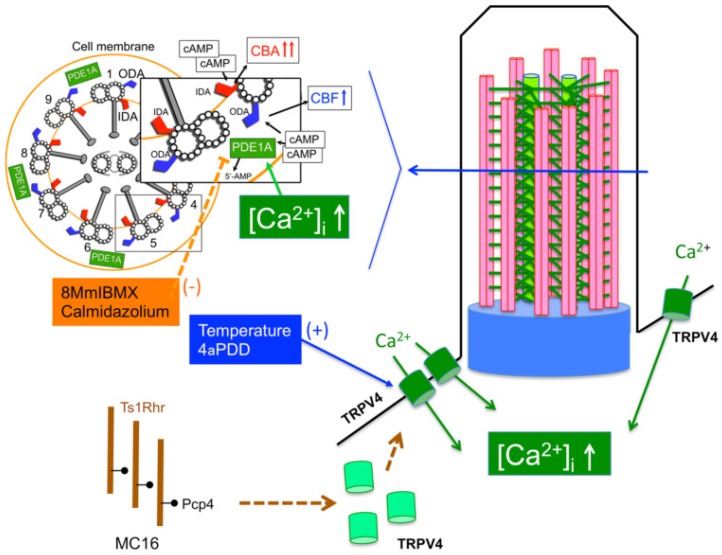
Schematic diagram of airway ciliary cells in DS. Three copies of *Pcp4* on Ts1Rhr induce overexpression of PCP4 leading to enhancement of Ca^2+^ entry via TRPV4 under the basal condition. A basal [Ca^2+^]_i_ increase on early embryonic day may affect axonemal regulatory complexes regulated by Ca^2+^. A basal [Ca^2+^]_i_ increase activates PDE1A existing in the metabolon regulating ODAs, leading to suppression of cAMP accumulation.

**Table 1 ijms-21-01947-t001:** CBFs and CBAs of airway cilia in Wt, Ts1Rhr, and Ts1Rhr:*Pcp4^+/+/-^*.

	CBF (Hz)	CBA (Degree)
Wt	11.1 ± 0.1 (*n* = 4)	93.1 ± 1.1 (*n* = 4)
Ts1Rhr	9.6 ± 0.5 (*n* = 4)	90.6 ± 0.6 (*n* = 4)
Ts1Rhr:*Pcp4^+/+/-^*	10.8 ± 0.2 (*n* = 3)	92.7 ± 0.1 (*n* = 3)

“n” shows the number of animals and values are means ± SD.

**Table 2 ijms-21-01947-t002:** CBF ratio and CBA ratio 5 min and 10 min after procaterol (Proc) stimulation.

**Proc (1 nM)**	**CBA Ratio**	**CBF Ratio (5 min)**	**CBF ratio (10 min)**
Wt	1.64 ± 0.01 (*n* = 5)	1.62 ± 0.01 (*n* = 5)	1.67 ± 0.02
Ts1Rhr	1.49 ± 0.04 (*n* = 5)	1.33 ± 0.04 (*n* = 6)	1.45 ± 0.04
Ts1Rhr:*Pcp4^+/+/-^*	1.44 ± 0.04 (*n* = 4)	1.42 ± 0.03 (*n* = 6)	1.48 ± 0.03
**Proc (10 nM)**	**CBA ratio**	**CBF ratio (5 min)**	**CBF ratio (10 min)**
Wt	1.73 ± 0.02 (*n* = 5)	2.04 ± 0.05 (*n* = 4)	1.96 ± 0.07
Ts1Rhr	1.59 ± 0.01 (*n* = 4)	2.07 ± 0.05 (*n* = 6)	2.00 ± 0.05

“n” shows the number of cells and values are means ± SEM.

**Table 3 ijms-21-01947-t003:** Time constant “τ” calculated from the fitted curve.

**Proc (1 nM)**	**τ_CBA_ (min)**	**τ_CBF_ (min)**
Wt	1.16 ± 0.04 (*n* = 5)	4.33 ± 0.43 (*n* = 5)
Ts1Rhr	1.16 ± 0.06 (*n* = 5)	* 6.05 ± 0.40 (*n* = 6)
Ts1Rhr:*Pcp4^+/+/-^*	1.16 ± 0.08 (*n* = 4)	† 4.39 ± 0.56 (*n* = 6)
**Proc (10 nM)**	**τ_CBA_ (min)**	**τ_CBF_ (min)**
Wt	1.46 ± 0.22 (*n* = 4)	2.63 ± 0.17 (*n* = 5)
Ts1Rhr	1.53 ± 0.07 (*n* = 4)	2.52 ± 0.12 (*n* = 6)
**8MmIBMX + Proc (1nM)**	**τ_CBA_ (min)**	**τ_CBF_ (min)**
Wt	2.73 ± 0.14 (*n* = 5)	2.85 ± 0.20 (*n* = 4)
Ts1Rhr	2.70 ± 0.09 (*n* = 5)	2.83 ± 0.21 (*n* = 5)
**Calmidaz + Proc (1 nM)**	**τ_CBA_ (min)**	**τ_CBF_ (min)**
Wt	3.21 ± 0.28 (*n* = 4)	3.23 ± 0.39 (*n* = 5)
Ts1Rhr	3.27 ± 0.39 (*n* = 4)	3.33 ± 0.55 (*n* = 5)

* or † significantly differerent from Wt or Ts1Rhr, respectively (*p* < 0.05). “n” shows the number of cells and values are means ± SEM.

**Table 4 ijms-21-01947-t004:** Effects of procaterol stimulation on CBF and CBA ratio in the presence of 8MmIBMX and calmidazolium (Calmidaz).

	CBA ratio		CBF Ratio	
**8MmIBMX (40 µM)**	Before Proc	Proc (5 min)	Before Proc	Proc (5 min)
**Wt**	1.21 ± 0.02 (*n* = 5)	1.67 ± 0.02	1.26 ± 0.03 (*n* = 4)	1.71 ± 0.02
**Ts1Rhr**	1.20 ± 0.02 (*n* = 5)	1.65 ± 0.02	1.20 ± 0.02 (*n* = 5)	1.65 ± 0.04
**Calmidaz (25 µM)**	Before Proc	Proc (5 min)	Before Proc	Proc (5 min)
**Wt**	1.13 ± 0.01 (*n* = 4)	1.45 ± 0.02	1.13 ± 0.02 (*n* = 5)	1.45 ± 0.02
**Ts1Rhr**	1.10 ± 0.01 (*n* = 4)	1.39 ± 0.02	1.40 ± 0.04 (*n* = 5)	1.65 ± 0.04

“n” shows the number of cells and values are means ± SEM.

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
