# Peer review of "Airway Ciliary Beating Affected by the *Pcp4* Dose-Dependent [Ca^2+^]_i_ Increase in Down Syndrome Mice, Ts1Rhr"

_ijms, 2020, doi:10.3390/ijms21061947_

Round 1
Reviewer 1 Report
The manuscript submitted by Kogiso and co-authors is an interesting work nicely complementing previous findings pointing to the significance of the PCP4 in ependymal cilia. This time Authors analyze the effect of the trisomy and a resumption to two copies of the Pcp4 gene on ciliary beat frequency and ciliary bend angle in airway cilia. The outcome is (as one could assume) similar as in case of ependymal cilia. Next, Authors investigate the effect of several factors on cilia beating in WT, Ts1Rh and Ts1Rhr:Pcp4+/+/- mice.
General suggetions:
- It would be really helpful to have all data from various experiments presented in one table (CBF, CBA, number of cells analyzed, number of animals). I think in this way data will be more clear , easier to follow and compare between experiments and the main text could me much shorter.
- Please consider providing movies showing cilia beating in airway cells in WT, Ts1Rh and Ts1Rhr:Pcp4+/+/- as supplementary data.
- Mutations in different ciliary genes affect CBF and CBA to a different extent. To allow a reader to form an opinion about the changes in cilia beating and put it into perspective of defects caused by mutations of other ciliary genes, could you please, in a discussion section, provide values of CBF and CBA in mice that are PCD models ?
- It would be really helpful to have a summary scheme showing the elements of the investigated pathway starting with Pcp4 and ending with CBF and CBA with marked calcium increase or decrease, cAMP, enzyme involved and used drugs.
- This manuscript should be read by a native speaker familiar with the subject discussed.
Minor comments:
Abstract:
Lines 21-24
Is: It has been demonstrated in Ts1Rhr, a mouse model of Down syndrome, that decreases in
basal ciliary beat frequency (CBF) and ciliary bend angle (CBA, an index of ciliary beat amplitude) occur in airway cilia, and that resumption to two copies of the Pcp4 gene on the Ts1Rhr trisomic segment (Ts1Rhr:Pcp4+/+/-) rescues the decreases in CBF and CBA that occur in Ts1Rhr.
Suggested:
In Ts1Rhr, a Down syndrome model mouse, the airway cilia beat with reduced basal ciliary beat frequency (CBF) and ciliary bend angle (CBA, an index of ciliary beat amplitude). A resumption to two copies of the Pcp4 gene on the Ts1Rhr trisomic segment (Ts1Rhr:Pcp4+/+/-) rescues the decreases in CBF and CBA that occur in Ts1Rhr.
Lines 25-27:
Is: Procaterol, a ß2-agonist, that stimulates a slower increase in CBF than in CBA via Ca2+/calmodulin-dependent phosphodiesterase 1 (PDE1) in airway cilia, stimulated a slower CBF increase in Ts1Rhr than in wild type mice (Wt) or in Ts1Rhr:Pcp4+/+/-.
Please rewrite to avoid this partial repetition
Line 31: Is: …in Wt or in Ts1Rhr:Pcp4+/+/-, suggesting the activation of PDE1A localizing airway cilia in Ts1Rhr.
Suggested:… in Wt or in Ts1Rhr:Pcp4+/+/-, suggesting the activation of PDE1A that is present in airway cilia in Ts1Rhr.
Introduction
Line 67 - Is: Ts1Rhr region hs also been demonstrated
Should be: Ts1Rhr region has also been demonstrated
Results
Lines 94-96 is: Figure 2A shows that the CBF and CBA of airway cilia increased by stimulation with 1 nM procaterol in Wt. Stimulation with procaterol increased CBF and CBA, but the time course of the
CBF increase was much slower than that of the CBA increase, as previously reported [18-21].
Suggested: In case of Wt, a stimulation with 1 nM procaterol causes an increase of CBF and CBA of airway cilia (Figure 2A) however, contrary to the previous reports, the time course of the CBF increase was much slower than that of the CBA increase [18-21].
Lines 127-128: These results suggest that procaterol stimulates a slower cAMP accumulation in the CBF-regulating metabolon in Ts1Rhr than in Wt or Ts1Rhr:Pcp4+/+/-.
Please, re-write
Line 166: please explain term ODA, and which parameter is altered by ODAs , cite REF about ODA and IDA role in cilia beating.
Methods line 342 – please indicate concentration before the name of the chemical e.g 121 (mM) NaCl.
Author Response
I have revised the manuscript according to comments of the reviewer 1.
General suggetions:
It would be really helpful to have all data from various experiments presented in one table (CBF, CBA, number of cells analyzed, number of animals). I think in this way data will be more clear, easier to follow and compare between experiments and the main text could me much shorter.
I added the Table 1, 2, and 4 (CBFs and CBAs of Wt, Ts1Rhr, and TS1Rhr:Pcp4+/+/-) and Table3 (tau of each experiment).
Please consider providing movies showing cilia beating in airway cells in WT, Ts1Rh and Ts1Rhr:Pcp4+/+/- as supplementary data.
I added the three movies of airway ciliary beating, Wt, Ts1Rhr, and Ts1Rhr:Pcp4+/+/- as supplemental files.
Mutations in different ciliary genes affect CBF and CBA to a different extent. To allow a reader to form an opinion about the changes in cilia beating and put it into perspective of defects caused by mutations of other ciliary genes, could you please, in a discussion section, provide values of CBF and CBA in mice that are PCD models ?
Most of PCD model mice shows no ciliary beating. I added this in the discussion. Line 308-314.
It would be really helpful to have a summary scheme showing the elements of the investigated pathway starting with Pcp4 and ending with CBF and CBA with marked calcium increase or decrease, cAMP, enzyme involved and used drugs.
I added the summary scheme, as Fig.8.
This manuscript should be read by a native speaker familiar with the subject discussed.
I sent his manuscript to AJE (a company for English check) before the submission. I enclosed the certificate from the AJE.
Minor comments:
Abstract:
Lines 21-24
Is: It has been demonstrated in Ts1Rhr, a mouse model of Down syndrome, that decreases in
basal ciliary beat frequency (CBF) and ciliary bend angle (CBA, an index of ciliary beat amplitude) occur in airway cilia, and that resumption to two copies of the Pcp4 gene on the Ts1Rhr trisomic segment (Ts1Rhr:Pcp4+/+/-) rescues the decreases in CBF and CBA that occur in Ts1Rhr.
Suggested:
In Ts1Rhr, a Down syndrome model mouse, the airway cilia beat with reduced basal ciliary beat frequency (CBF) and ciliary bend angle (CBA, an index of ciliary beat amplitude). A resumption to two copies of the Pcp4 gene on the Ts1Rhr trisomic segment (Ts1Rhr:Pcp4+/+/-) rescues the decreases in CBF and CBA that occur in Ts1Rhr.
I have changed the this part as follows.
In Ts1Rhr, a Down syndrome model mouse, the airway ciliary beatings are impaired; that is decreases in ciliary beat frequency (CBF) and ciliary bend angle (CBA, an index of ciliary beat amplitude)). A resumption to two copies of the Pcp4 gene on the Ts1Rhr trisomic segment (Ts1Rhr:Pcp4+/+/-) rescues the decreases in CBF and CBA that occur in Ts1Rhr.
Lines 25-27:
Is: Procaterol, a ß2-agonist, that stimulates a slower increase in CBF than in CBA via Ca2+/calmodulin-dependent phosphodiesterase 1 (PDE1) in airway cilia, stimulated a slower CBF increase in Ts1Rhr than in wild type mice (Wt) or in Ts1Rhr:Pcp4+/+/-.
Please rewrite to avoid this partial repetition
I rewrote this part as follows.
Procaterol, a ß2-agonist, that stimulates a slower increase in CBF than in CBA via Ca2+/calmodulin-dependent phosphodiesterase 1 (PDE1) in airway cilia of wild type mice (Wt), also increased CBF and CBA in Ts1Rhr, but the time course of CBF increase was much slower than that of Wt or Ts1Rhr:Pcp4+/+/-.
Line 31: Is: …in Wt or in Ts1Rhr:Pcp4+/+/-, suggesting the activation of PDE1A localizing airway cilia in Ts1Rhr.
Suggested:… in Wt or in Ts1Rhr:Pcp4+/+/-, suggesting the activation of PDE1A that is present in airway cilia in Ts1Rhr.
I changed as follows.
Measurements of cAMP revealed that the cAMP contents were lower in Ts1Rhr than in Wt or in Ts1Rhr:Pcp4+/+/-, suggesting the activation of PDE1A that is present in airway cilia in Ts1Rhr.
Introduction
Line 67 - Is: Ts1Rhr region hs also been demonstrated
Should be: Ts1Rhr region has also been demonstrated
i changed to "has".
Results
Lines 94-96 is: Figure 2A shows that the CBF and CBA of airway cilia increased by stimulation with 1 nM procaterol in Wt. Stimulation with procaterol increased CBF and CBA, but the time course of the CBF increase was much slower than that of the CBA increase, as previously reported [18-21].
Suggested: In case of Wt, a stimulation with 1 nM procaterol causes an increase of CBF and CBA of airway cilia (Figure 2A) however, contrary to the previous reports, the time course of the CBF increase was much slower than that of the CBA increase [18-21].
I have changed as follows (L96-99).
Figure 2A shows that the CBF and CBA of airway cilia increased by stimulation with 1 nM procaterol in Wt. Stimulation with procaterol increased CBF and CBA, but the time course of the CBF increase was much slower than that of the CBA increase. Similar results have already shown in previously reports [18-21].
Lines 127-128: These results suggest that procaterol stimulates a slower cAMP accumulation in the CBF-regulating metabolon in Ts1Rhr than in Wt or Ts1Rhr:Pcp4+/+/-.
Please, re-write
I changed to follows (Line 147).
These results suggest that accumulations of cAMP in the metabolon regulating ODAs are slow in Ts1Rhr compared with those in Wt and Ts1Rhr:Pcp4+/+/-.
Line 166: please explain term ODA, and which parameter is altered by ODAs , cite REF about ODA and IDA role in cilia beating.
I added followings (Line 141-147)
Previous studies in Chlamydomonas and Tetrahymena mutants and in primary ciliary dyskinesia (PCD) showed that the outer dynein arms (ODAs) and inner dynein arms (IDAs) are functionally distinct: the ODAs control CBF, and the IDAs control the waveform including CBA [22-25]. Moreover, during procaterol stimulation, the slower CBF increase than CBA increase has been shown to be caused by Ca2+/calmodulin dependent phosphodiesterase 1A (PDE1A), which exists in the metabolon regulating ODAs (outer dynein arms) and degrades cAMP dependent on [Ca2+]i [19-21].
Methods line 342 – please indicate concentration before the name of the chemical e.g 121 (mM) NaCl.
I have changed according to the reviewers suggestion.
Reviewer 2 Report
The manuscript “Airway ciliary beating affected by the Pcp4 dose-dependent [Ca2+]i in
Down syndrome mice, Ts1Rhr”, investigates the role of Pcp4 (Purkinje cell protein 4) in CBF and CBA decrease in Ts1Rhr DS mouse model.
The introduction is relevant and theory based. Sufficient information about previous findings is presented for lay readers to follow the present study rationale and procedures. Methods are appropriate.
Comments and Suggestion for Authors
Please describe how and why the doses of calmidazolium 25 uM and 8MmIBMX 40 μM were chosen for experiments.
I wonder if this findings may be usefull for development of therapeutic strategies in the future.
Author Response
Reviewer #2
Comments and Suggestion for Authors
Please describe how and why the doses of calmidazolium 25 uM and 8MmIBMX 40 μM were chosen for experiments.
I added the following sentence (Line 175 and Line 201).
Line 175: A previous study demonstrated that 8MmIBMX at 40 µM inhibits PDE1A in airway ciliary cells [19-21].
Line 201: Because PDE1 is a Ca2+/calmodulin dependent enzyme and calmidazolium at 25 µM has been shown to inhibit PDE1A in airway ciliary cells [19-21].
I wonder if this findings may be usefull for development of therapeutic strategies in the future.
I agree with reviewers comments.
I changed the last paragraph of the Discussion (Line 322-327)
In DS children, respiratory diseases, such as bronchopneumonia, are the main cause of death [5,6], and they suffer from recurrent and prolonged upper respiratory infections. DS children have congenital abnormalities of their airways, heart and immune system, which induce frequent respiratory problems, such as bronchopneumonia. Moreover, dysfunction of ciliary beating also induces recurrent and prolonged infections of airways and tends to be worse in lung diseases compared with that in typical patients [9,35,36].
Round 2
Reviewer 1 Report
Minor corrections suggested:
lines 24-27 - please divide this sentence.
line 60 it seems that the word" harbors" is missing
line 98: is "Experiments were carried.." suggested change : "Next the experiments...
line 147 - is themetabolon - change to the metabolon
Table 2 from the new line
line 269 is "On the other hand The ODA and IDA" - suggested: start with a new line and remove "on the other hand"
line 301: cause caused
Please add legend to the Fig 8 and briefly describe the presented pathway.
Author Response
I have revised the manuscript "ijms-730895"
The responses to the reviewer's comments are follows.
Minor corrections suggested:
lines 24-27 - please divide this sentence.
In airway cilia, upon stimulation with procaterol (a ß2-agonist), the CBF increase is slower over the time course than the CBA increase because of cAMP degradation by Ca2+/calmodulin-dependent phosphodiesterase 1 (PDE1) existing in the metabolon regulating CBF. In Ts1Rhr, procaterol-stimulated CBF increase was much slower over the time course than in the wild type mouse (Wt) or Ts1Rhr:Pcp4+/+/-.
line 60 it seems that the word" harbors" is missing
Ts65DN, which harbors ~86 protein coding genes from miR155 to Znf295 is a large trisomy, Ts1Cje, which harbors ~66 protein coding genes from Sfrs15 to Znf295 is shorter but overlaps with Ts65DN.
line 98: is "Experiments were carried.." suggested change : "Next the experiments...
Next, the same experiments were carried out using Ts1Rhr (Fig. 2B).
line 147 - is themetabolon - change to the metabolon
which exists in the metabolon regulating ODAs (outer dynein arms) and degrades cAMP dependent on [Ca2+]i[19-21].
Table 2 from the new line
suggesting a high [Ca2+]iin the metabolon regulating ODAs.
Table 2.CBF ratio and CBA ratio 5 min and 10 min after procaterol (Proc) stimulation.
line 269 is "On the other hand The ODA and IDA" - suggested: start with a new line and remove "on the other hand"
Previous studies have shown that airway ciliary cells express TRPV4 channels activated by temperature or 4aPDD, and that the TRPV4 channels participate in the maintenance of the basal [Ca2+]iin airway ciliary cells [26,27].
The ODAs and IDAs are different in their functions: the ODAs control CBF, and the IDAs control the waveform including CBA [22-25].
line 301: cause caused
mutation of the outer dynein arm heavy chain 11 locus caused no ciliary beating [38].
Please add legend to the Fig 8 and briefly describe the presented pathway.
Figure 8.Schematic diagram of airway ciliary cells in DS. Three copies of Pcp4 on Ts1Rhr induce overexpression of PCP4 leading to enhancement of Ca2+ entry via TRPV4 under the basal condition. A basal [Ca2+]I increase on early embryonic day may affectaxonemal regulatory complexes regulated by Ca2+. A basal [Ca2+]I increase activates PDE1A existing in the metabolon regulating ODAs,leading to suppression of cAMP accumulation.
I changed the final paragraph of the discussion (Line 324-327)
The result of this study is summarized in Fig 8. Three copies of Pcp4on Ts1Rhr stimulate TRPV4 expression leading to increase basal [Ca2+]i. A basal [Ca2+]iincrease on early embryonic days may affect axonemal regulatory complexes regulated by Ca2+signals and also it affects airway ciliary beating via interactions between cAMP and Ca2+signals, such as PDE1A activation in Ts1Rhr.